# Characteristics of creative individuals: An umbrella review protocol

**Alex Thabane**[1]*, **Tyler McKechnie**[1,2], **Phillip Staibano**[1,2], **Goran Calic**[3], **Colin Kruse**[2], **Jason W. Busse**[1,4], **Sameer Parpia**[1], **Mohit Bhandari**[1,2]

**1** Department of Health Research Methods, Evidence, and Impact, McMaster University, Hamilton, Ontario, Canada, **2** Department of Surgery, McMaster University Medical Center, Hamilton, Ontario, Canada, **3** DeGroote School of Business, McMaster University, Hamilton, Ontario, Canada, **4** Department of Anesthesia, McMaster University Medical Center, Hamilton, Ontario, Canada

* thabanla@mcmaster.ca

**Data Availability Statement:** No datasets were generated or analysed during the current study. All relevant data from this study will be made available upon study completion.

## Abstract

### Introduction

The properties of creative products–novelty and usefulness–are generally agreed upon by researchers. Yet, consensus is lacking on which personal and environmental factors contribute to an individual's creative potential, or to what extent. substantial research work has been conducted in this area, leading to the publication of many systematic reviews and meta-analyses collating the available evidence. However, many of these reviews have differing methodological and theoretical characteristics, and often report conflicting results. To summarize the current review literature on factors associated with creativity and better understand the similarities and discrepancies among reviews on the same topic, we plan to conduct an umbrella review of reviews.

### Methods

This protocol has been registered in Open Science Framework (DOI: 10.17605/OSF.IO/H78YS). We will search MEDLINE, PsycINFO, Web of Science, EMBASE, and Google Scholar for peer-reviewed reviews exploring factors associated with creativity. The quality of the included reviews will be assessed using the Risk of Bias Assessment Tool for Systematic Reviews (ROBIS), and the degree of study overlap across reviews assessed through calculation of the corrected covered area (CCA). Descriptive analyses will be performed to synthesize the results of the included reviews. We plan to use the results to create a holistic framework of the factors associated with creativity, which could be used to cross-validate existing creativity theories, or create new ones.

### Ethics and dissemination

The results of this umbrella review will be published in a relevant psychology-focused journal and presented at local, national and international conferences. As all data is peer-reviewed and presented in aggregate, we will not require ethics approval.

**Funding:** The author(s) received no specific funding for this work.

**Competing interests:** The authors have declared that no competing interests exist.

## Introduction

While Van Gogh's *The Starry Night* or da Vinci's *Mona Lisa* are undeniably creative, the factors which facilitate these creative products are much more contested. The importance of identifying factors contributing to creative potential is not a trivial matter: the measurement, cultivation, and broader study of creativity is dependent on a theoretical framework that is accurate, comprehensive, and valid.

Extensive research on factors associated with creativity (e.g., intelligence, intrinsic motivation, personality factors, environmental features) has been conducted by the scientific community [1–5]. This multifactorial conceptualization of creativity is reflected in the most recent theories of creativity, such as the Investment Theory of Creativity [6], Componential Theory of Creativity [7], and Systems Theory of Creativity [8]. The volume of literature in this area has led to the publication of many systematic reviews and meta-analyses looking to collate the existing evidence [9–16]. Yet, there is still a lack of consensus and understanding on which factors are associated with creativity. For instance, intelligence was found to have a negligible correlation with creativity test scores in an early meta-analysis [9], yet subsequent meta-analyses have found that various aspects of intelligence correlate with creativity [17, 18]. To date, there is still debate about if, or to what extent, intelligence is a requisite of creativity [2, 18–21]. Similarly, differing levels of correlation with creativity have been reported in meta-analyses exploring personality factors, motivation, and environment [11, 14–16, 22].

The existence of a large body of systematic review and meta-analyses permits the conduct of an umbrella review, or 'overview of reviews'. This has yet to be performed; we found only one review of meta-analyses, which was limited to only 7 studies and synonymized creativity and innovation [23]. Thus, we propose a comprehensive umbrella review to integrate the current review literature on factors associated with creativity, identify similarities and discrepancies in results across reviews, and explore the possible reasons for such discrepancies. Discrepancies among reviews is not likely to be solely due to the availability of new primary research over time: the introduction of more robust creativity measures and testing practices, use of outdated intelligence tests, and narrow definition of creativity informing study inclusion criteria are just some methodological and theoretical factors that may explain the differing conclusions across reviews [24–26]. This umbrella review will not only contribute to consensus generation on factors associated with creativity but improve the understanding of the effects of various methodological and theoretical factors on creativity research findings.

## Methodology

This protocol was registered on the Open Science Framework (DOI: 10.17605/OSF.IO/H78YS) and designed in accordance with the Joanna Briggs Institute guidelines for the conduct of umbrella reviews [27]. While there is no specific guideline for the reporting of umbrella review protocols, this protocol was reported in accordance with the Preferred Reporting Items for Systematic review and Meta-Analysis Protocols (PRISMA-P) statement (**S1 File**) [28].

## Objectives

Our primary objective is to explore the existing systematic review literature on the relationship between creativity and the following factors: sex; sociocultural background; intelligence; personality; motivation; mood; psychopathology; environment.

## Theoretical assumptions

Underlying the rationale for this umbrella review is the assumption that individual creativity is multifactorial in nature, requiring aspects including but not limited to intelligence, knowledge, cognition, personality, motivation, and affect, and is influenced by the environmental context [29]. This conceptualization is in alignment with the definition of creativity as the "interaction between aptitude, process, and environment by which an individual or group produces a perceptible product that is both novel and useful as defined within a social context", which we plan to use as a theoretical basis for determining study eligibility and interpreting the results [30]. However, definitions of creativity are highly variable [31], and assessment methods equally so [32, 33]. We will thus include reviews exploring creativity as defined by definitions other than the one we have presented here.

## Eligibility criteria

Articles eligible for inclusion in this umbrella review will have met all of the following criteria:

1. Systematic review (with or without meta-analysis);

2. Empirically investigated the relationship between creativity and one or more personal and/or environmental factors;

3. Included primary research studies conducted in humans; and

4. Published in a peer-reviewed journal

We will exclude dissertations, pre-prints, and conference abstracts. Studies published in non-English languages will be translated with Google Translate.

## Information sources and search strategy

We plan to systematically search MEDLINE, PsycINFO, Web of Science, and EMBASE databases from inception to 1 September 2024. Additionally, we will search Google Scholar for eligible reviews, and use an ascendancy approach to identify additional articles. As an example, we developed the search strategy (**S2 File**) for PsycINFO using a combination of keywords and subject headings, combined with Boolean operators (AND; OR). For each of the remaining databases, a search strategy will be developed tailored to the index structure of each database.

## Study screening

We plan to upload the results of our searches to Covidence, which removes duplicates and provides a platform for screening, full-text review, and data extraction. Title and abstract screening, and subsequently full-text review, will be performed independently and in duplicate by a pair of reviewers, with discrepancies resolved by discussion to achieve consensus. If necessary, a third reviewer will act as adjudicator. The results of our study screening procedure will be presented using a PRISMA flow chart.

## Data extraction

Using the Covidence platform, a standardized template will be developed to extract data from eligible studies. A pair of reviewers will independently extract the data, with any discrepancies resolved by discussion to achieve consensus, or adjudication by a third reviewer. The following review-level data will be extracted: first author; year of publication; journal name; country of conduct; number of primary studies included; total sample size of participants across primary studies; personal and/or environmental factors explored (e.g., personality, intrinsic

motivation); the pooled effect size metric (e.g., Pearsons $r$) with the 95% confidence interval [95% CI] and associated heterogeneity statistics (e.g., Cochran's Q test, $I^2$) for each factor; and whether meta-analysis method was conducted using fixed effects or random effects models. For each primary study within each review, the following data will be extracted: first author; year of publication; sample size; population description; factor(s) explored; tool used to assess creativity; tools used to measure factor(s); and effect estimate with the associated 95% CI. In the event of missing data, efforts will be made to contact the first author to acquire the data.

### Risk of bias assessment

We will use the Risk of Bias Assessment Tool for Systematic Reviews (ROBIS) to assess the quality of included systematic reviews [34]. The ROBIS tool assess the risk of bias in systematic reviews from 4 different domains: study eligibility criteria; identification and selection of studies, data collection and study appraisal; and synthesis of findings. The results of the risk of bias assessment will be reported narratively and tabulated in accordance with the recommendations of Whiting et. al. [34].

### Data synthesis and statistical analysis

We plan to conduct descriptive analyses to synthesize the findings across reviews. To begin, review characteristics, including the various methodological and theoretical features of each review, will be described, and the results from each individual review reported and tabulated, indicating the strength and direction of the relationship. Then, we will narratively synthesize the data across all reviews, organized by factor, comparing the results across reviews, changes in results over time, and differences in methodological and theoretical approaches between reviews. The degree of study overlap in review on the same topic will be assessed by calculating the corrected covered area (CCA); CCA values between 0 and 5 will be considered slight overlap, 6–10 considered moderate overlap, 11–15 considered high overlap, and >15 considered very high overlap [35].

To summarize the findings of the umbrella review and provide a cohesive model to explain the results across reviews, we will create a conceptual framework to illustrate the magnitude and direction of association between the explored factors and creativity.

### Timeline

The planned start date for this study, beginning with the literature search, is scheduled for November 2024. A first draft is planned for May 2025, with a final draft ready for submission to a journal by July 2025.

### Study limitations

Anticipated limitations include variable study quality and differing methodological and theoretical characteristics of the included reviews. To address these limitations we plan to assess the study quality of the included reviews using the ROBIS tool [34], and will compare the results between studies with differing methodological and theoretical characteristics. We also expect there to be significant overlap in included studies across reviews of the same topic. We plan to calculate the degree of study overlap using the CCA metric.

## Ethics and dissemination

### Ethical considerations

As this review includes published studies and presents aggregated data, no ethics approval is required for this study.

### Knowledge translation

The results of this umbrella review will be published in a creativity-specific peer-reviewed journal. Efforts will be made to disseminate the results of the study to relevant stakeholders, including creativity researchers and psychometricians. Following the publication of the review, the lead author plans to present the results of this study at local, national, and international psychology conferences.

## Conclusion

A large body of systematic review and meta-analysis literature exists on the personal and environmental factors associated with creativity. A synthesis of this body of evidence, in the form of an umbrella review, will holistically assess the factors associated with individual creativity, as well as explore the similarities and differences in results across reviews on the same topics. The results of this umbrella review will also contribute an important conceptual framework which could be used to validate existing theories and measurement tools of creativity, create new ones.

## Supporting information

**S1 File. PRISMA-P checklist.**
(DOCX)

**S2 File. Search strategy for PsycINFO.**
(DOCX)

## Author Contributions

**Conceptualization:** Alex Thabane.

**Methodology:** Alex Thabane, Tyler McKechnie, Phillip Staibano, Goran Calic, Colin Kruse, Jason W. Busse, Sameer Parpia, Mohit Bhandari.

**Supervision:** Mohit Bhandari.

**Writing – original draft:** Alex Thabane.

**Writing – review & editing:** Alex Thabane, Tyler McKechnie, Phillip Staibano, Goran Calic, Colin Kruse, Jason W. Busse, Sameer Parpia, Mohit Bhandari.

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
