## [Decision Letter · Decision Letter 0]

25 Sep 2024

Characteristics of Creative Individuals: An Umbrella Review Protocol

PONE-D-24-35304

Dear Dr. Thabane,

We’re pleased to inform you that your manuscript has been judged scientifically suitable for publication and will be formally accepted for publication once it meets all outstanding technical requirements. Although Reviewer 2 suggested minor revisions, we believe these changes do not warrant another round of review, as they do not compromise the understanding or the results of your work.

Kind regards,

André Ramalho, PhD

Academic Editor

PLOS ONE

Reviewers' comments:

Reviewer's Responses to Questions

**Comments to the Author**

1. Does the manuscript provide a valid rationale for the proposed study, with clearly identified and justified research questions?

Reviewer #1: Yes

Reviewer #2: Yes

2. Is the protocol technically sound and planned in a manner that will lead to a meaningful outcome and allow testing the stated hypotheses?

Reviewer #1: Yes

Reviewer #2: Partly

3. Is the methodology feasible and described in sufficient detail to allow the work to be replicable?

Reviewer #1: Yes

Reviewer #2: Yes

4. Have the authors described where all data underlying the findings will be made available when the study is complete?

Reviewer #1: Yes

Reviewer #2: Yes

5. Is the manuscript presented in an intelligible fashion and written in standard English?

Reviewer #1: Yes

Reviewer #2: No

6. Review Comments to the Author

You may also provide optional suggestions and comments to authors that they might find helpful in planning their study.

Reviewer #1: - This study protocol proves to be interesting and relevant in theme and scope considering consensus is lacking on wich factors contribute to an individual´s creative potential, or to what extent. Besides, the existing literature presents conflicting results. A study that summarizes the current literature on this same topic is of a great importance.

- By registering this protocol on the Open Science Framework, designing it in accordance with the Joanna Briggs Institute guidelines for the conduct of umbrella reviews, and reporting it in accordance with PRISMA-P statement, authors present a feasible methodology which allows the work replicability, thus leading to a meaningful outcome.

- It would be important if authors could share in the Umbrella review the final queries applied in each database screended in the study, aiming to ensure the transparency of the applied methods, and permitting a user to address the same question and screen the same set of literature to come up with a comparable general conclusion.

Given all these aspects, it is believed this manuscript is suitable for publication.

Best regards.

Reviewer #2: This is an interesting protocol, but it has a few shortcomings.

1. The methodology, especially the inclusion-exclusion criteria, should be defined with more clarity

2. The methodology using a diagram or figure would be more helpful for the audience.

3. The term "umbrella review" should be defined for the audience.

4. The purpose or societal impact of such a review should have been defined in detail.

7. PLOS authors have the option to publish the peer review history of their article (what does this mean?). If published, this will include your full peer review and any attached files.

Reviewer #1: **Yes: **Bruno Filipe Coelho da Costa

Reviewer #2: **Yes: **Dr Anamika

---

## [Editor Report · Acceptance letter]

30 Sep 2024

PONE-D-24-35304 

PLOS ONE

Dear Dr. Thabane, 

I'm pleased to inform you that your manuscript has been deemed suitable for publication in PLOS ONE. Congratulations! Your manuscript is now being handed over to our production team.

Kind regards, 

on behalf of

Prof. Dr. André Ramalho 

Academic Editor

PLOS ONE